# Cluster randomised controlled trial to determine the effect of peer delivery HIV self-testing to support linkage to HIV prevention among young women in rural KwaZulu-Natal, South Africa: a study protocol

Oluwafemi Atanda Adeagbo [1,2] Nondumiso Mthiyane,[1] Carina Herbst,[1] Paul Mee,[3] Melissa Neuman,[3] Jaco Dreyer,[1] Natsayi Chimbindi,[1] Theresa Smit,[1] Nonhlanhla Okesola,[1] Cheryl Johnson,[4] Karin Hatzold,[5] Janet Seeley,[1,6] Frances Cowan,[7,8] Liz Corbett,[9] Maryam Shahmanesh[1,10]

For numbered affiliations see end of article.

**Correspondence to**
Dr Maryam Shahmanesh;
m.shahmanesh@ucl.ac.uk

## ABSTRACT

**Introduction** A cluster randomised controlled trial (cRCT) to determine whether HIV self-testing (HIVST) delivered by peers either directly or through incentivised peer-networks, could increase the uptake of antiretroviral therapy and pre-exposure prophylaxis (PrEP) among young women (18 to 24 years) is being undertaken in an HIV hyperendemic area in KwaZulu-Natal, South Africa.

**Methods and analysis** A three-arm cRCT started mid-March 2019, in 24 areas in rural KwaZulu-Natal. Twenty-four pairs of peer navigators working with ~12 000 young people aged 18 to 30 years over a period of 6 months were randomised to: (1) *incentivised-peer-networks:* peer-navigators recruited participants 'seeds' to distribute up to five HIVST packs and HIV prevention information to peers within their social networks. Seeds receive an incentive (20 Rand = US$1.5) for each respondent who contacts a peer-navigator for additional HIVST packs to distribute; (2) *peer-navigator-distribution:* peer-navigators distribute HIVST packs and information directly to young people; (3) *standard of care:* peer-navigators distribute referral slips and information. All arms promote sexual health information and provide barcoded clinic referral slips to facilitate linkage to HIV testing, prevention and care services. The primary outcome is the difference in linkage rate between arms, defined as the number of women (18 to 24 years) per peer-navigators month of outreach work (/pnm) who linked to clinic-based PrEP eligibility screening or started antiretroviral, based on HIV-status, within 90 days of receiving the clinic referral slip.

**Ethics and dissemination** This study was approved by the Institutional Review Boards at the WHO, Switzerland (Protocol ID: STAR CRT, South Africa), London School of Hygiene and Tropical Medicine, UK (Reference: 15 990–1), University of KwaZulu-Natal (BFC311/18) and the KwaZulu-Natal Department of Health (Reference: KZ_201901_012), South Africa. The findings of this trial will be disseminated at local, regional and international meetings and through peer-reviewed publications.

### Strengths and limitations of this study

► There is no evidence on the use of HIV self-testing to improve linkage to effective HIV prevention such as pre-exposure prevention.

► There is limited evidence on the strength and limitations of different peer-to-peer approaches to improve uptake of HIV testing and linkage to care and prevention.

► Strengths include the use of a cluster randomised controlled trial (cRCT) with rigorous measurement of the outcome linkage to care or prevention by arm, combined with process evaluation and cost-effectiveness studies.

► By embedding this cRCT within a longitudinal demographic surveillance setting, we are able to measure the population reach of the intervention.

► Limitations include a small risk of contamination across clusters and potential for coercive test or intimate partner violence.

**Trial registration number** NCT03751826; Pre-results.

## INTRODUCTION

South Africa has the largest burden of HIV globally with 14% national prevalence rate and an estimated 7.9 million people living with HIV in 2017.[1] The province of KwaZulu-Natal (KZN) is mostly affected by the epidemic with an 18.1% prevalence rate in 2017,[1] while our research setting in uMkhanyakude district has an estimated 30% in the general population.[2] Of the new 88 000 HIV infections recorded among young people aged 15 to 24 years in 2017, 66 000 were among females.[1] Similarly, there is

high HIV incidence rate in adolescent girls and young women (AGYW) with an estimated 5% per annum in aged 15 to 19 years and 8% per annum in aged 20 to 24 years, respectively, in our research setting in Hlabisa subdistrict.[3] This high incidence persists despite an increasing range of effective HIV prevention and treatment interventions, including condoms, antiretroviral (ART) based prevention for example, pre-exposure prophylaxis (PrEP), universal test and treat[4 5] and voluntary medical male circumcision (VMMC).

Evidence from South Africa and other countries in sub-Saharan Africa shows that this is partly due to the fact that many young people living with HIV are undiagnosed and therefore not linked to care.[6 7] Similarly, a recent treatment-as-prevention trial conducted in this area failed to show an impact on incidence in part due to the challenge of testing and linking young women.[8] Patient level fears (eg, stigma, labelling and discrimination) and facility level barriers (eg, distance, waiting times and provider attitudes) continue to be barriers to young people not seeking HIV care services (HIV testing and uptake and adherence to antiretroviral therapy for treatment) in health facilities.[9–11] There is an urgent need to increase the proportion of those (particularly AGYW) who know their HIV status and take up effective HIV treatment as well as prevention — including ART based care and prevention.

To increase global testing rates and early access to treatment or PrEP, HIV self-testing (HIVST) — a simple saliva or blood-based self-test similar to a pregnancy test — has been identified as a potential method given its privacy and convenience.[12–16] Studies from different countries including South Africa have shown high acceptability and uptake of HIVST particularly among first time testers and young people.[13 14 16–19] Also, a growing number of studies have shown that rapid oral fluid testing was preferred to blood-based testing.[20–22] The OraQuick In-home HIV test (OraSure Technologies, Inc, Bethlehem, Pennsylvania, manufactured in Thailand) was recently pre-qualified by the WHO for international procurement.[23] Field based use confirmed the high accuracy of HIVST, although with some variability across different educational levels.[14 18 24] HIVST (OraQuick) product is currently available in South Africa and has been endorsed by the National Department of Health (NDoH) in those aged 18 and above, with recommendations emphasising the need for healthcare worker supported testing in those aged 18 and under.

Effective biomedical innovations such as PrEP have the potential to be gamechangers in the HIV epidemic in South Africa as part of the combination HIV prevention strategy and have thus been recommended for key populations such as sex workers, men who have sex with men and adolescent girls and young women aged 15 to 24 by the NDoH in South Africa.[25] However, their effectiveness will depend on HIV testing uptake and subsequent linkage to care and prevention.[26–28] The key

findings from systematic reviews of the HIV treatment cascade suggest that: (1) community-based delivery models, including adherence clubs, community health workers delivering de-centralised care and task-shifting to lay caregivers providing support across conditions, improve both ART uptake and sustained retention in low- and middle-income settings[29–31]; (2) peer support is effective to deliver health intervention particularly to hard-to-reach groups.[32 33] Moreover, there is some evidence to suggest that HIVST can improve linkage to treatment when coupled with community based support.[19 34] However, there is limited evidence of the effectiveness of HIVST to link people who are negative to prevention, and in particular PrEP, with or without community-based support.

Here, we describe a cluster randomised controlled trial to address this critical gap in HIVST evidence and linkage for young women aged 18 to 24 years. Although all young people aged 18 to 30 years are included in the peer-led community based promotion of HIV testing and linkage to HIV prevention and care, the aim of this trial is to determine whether HIVST delivered by peers either directly or through incentivised peer-networks, can increase the uptake of ART and PrEP among adolescent girls and young women (18 to 24 years) in a high HIV transmission setting in KwaZulu-Natal, South Africa. To the best of our knowledge, this is one of the first trials to test the effectiveness of oral-based HIVST to improve uptake of prevention in South Africa.

## STUDY AIMS AND OBJECTIVES

The specific aims of this trial are to: (1) increase the knowledge of HIV status among young women aged 18 to 24 years old and their young male partners through the distribution of HIVST through incentivised peer networks or direct distribution by peer navigators compared with peer navigators referring them into HIV testing services; (2) determine an increase in the rate of linkage among young women aged 18 to 24 years to HIV prevention and treatment services facilitated by distribution of HIVST through incentivised peer networks or direct distribution by peer navigators compared with peer navigators referring into services; (3) determine an overall increase in young men and women aged 18 to 30 aware of their status and linked to HIV care and prevention; (4) conduct a process evaluation of the acceptability, feasibility and reach (out of school, recently migrant and living in remote areas) in linking 18- to 24-year-old women to HIV prevention and treatment services of HIVST distribution through incentivised peer networks, or direct distribution by peer navigators or peer navigators referring into services and (5) measure the cost per 18- to 24-year-old woman linked to prevention and care through peer-led incentivised HIVST delivery system or direct distribution of HIVST by peer navigators, compared with peer navigator referring them to services.

## METHODS/DESIGN

This is a three-arm cluster randomised controlled trial (two intervention arms and one control arm) launched in mid-March 2019 and currently being carried out by 24 pairs of peer navigators that have been randomly assigned to one of three arms. The following subheadings grapple with the methods, outcomes, procedures and study design among others.

### Study setting and population

This study will be conducted in Africa Health Research Institute's (AHRI) longstanding demographic surveillance area in northern KwaZulu-Natal. The study area is mostly rural, and poor compared with other parts of South Africa, with high levels of unemployment (over 85% of young adults aged 20 to 24 years are unemployed) and the local language is isiZulu.[8] In the study area, 8 out of 100 women aged between 20 and 24 years acquire HIV in 1 year, and 4 out of 10 women attending antenatal clinics are found to be infected with HIV. Data between 2011 and 2015 in the study area suggests that sexually active women aged 16 to 29 and young adult men have an HIV incidence above the threshold of eligibility for PrEP.

The demographic surveillance area provides over 16 years of household history, and over a million person-years of follow-up through annual individual-level surveys, which capture sexual behaviour and partnerships, reproductive histories and contraception use, access to HIV testing and care, access to HIV prevention services (including VMMC), as well as socio-demographic information. Moreover, through a Memorandum of Understanding with the Department of Health, AHRI has also embedded data collection clerks within the public health clinics to capture electronically any clinical attendance and linking it with the surveillance platform on all consenting attendees. This allows us to measure linkage of individuals to HIV care and use of contraceptive services. As part of a US National Institute of Health (NIH) R01 we have selected, trained and employed 24 pairs of peer navigators, working in 24 discrete areas (based on administrative divisions) of the demographic surveillance area (the Hlabisa district of uMkhanyakude district of northern KwaZulu-Natal, South Africa) to deliver HIV and sexual health related promotion to an estimated 12 000 youth (male and female) aged 18 to 30 years (~500 per each of the 24 administrative areas) and young women aged 18 to 24 years residing in the administrative areas (figure 1).

### Theory of change

The intervention that is being tested in this cluster randomised controlled trial (cRCT) is guided by a theory of change developed through mental models and deductive development[35] entrenched in ecological approach.[36] We theorised that the distribution of HIVST kits (including linkage information and referral slips) via peer navigators or peer social networks (respondent driven sampling - RDS) would lead to improved HIV prevention cascade, HIV testing uptake and linkage to HIV treatment or prevention

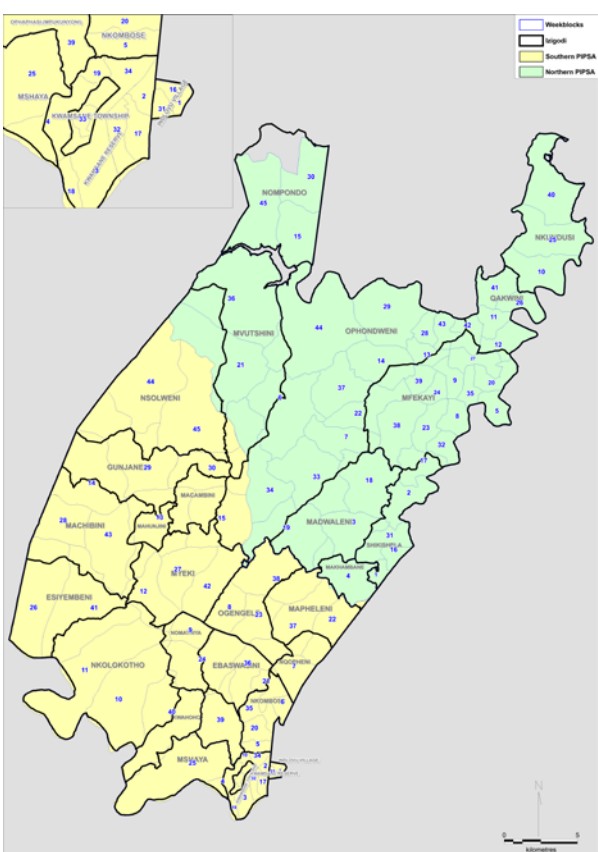

**Figure 1** Map of study sites in Hlabisa subdistrict in KwaZulu-Natal, South Africa.

services such as PrEP, among young women aged 18 to 24 years by creating peer-led demand, supporting young people to explore their candidacy for HIV care and prevention in privacy, and using social networks to reach those who need it most.[37] Work done by our group suggested that various factors associated with 'ecological framework' such as the fear of HIV-related stigma of attending a clinic for HIV testing and discrimination from healthcare providers or community may be addressed by HIVST since individuals can test privately anywhere without fear of being seen or judged.[2 38] Furthermore, formative work from our group suggested that community based delivery of services through youth friendly and accessible clinics for the study participants (walk-ins and those who present the study referral slips) could provide confirmatory HIV testing, treatment, prevention, contraceptives and other health services.[39 40] Following this, we developed a peer-to-peer intervention to reduce the burden of HIV among young women. We used the Standard Protocol Items: Recommendations for Interventional Trials reporting guidelines in this article.[41]

### Trial design

This cRCT is comparing two models of peer delivery of HIVST in the study sites through incentivised respondent driven peer networks and direct distribution by peer navigators compared with standard of care (referral to HIV testing, prevention and care services by peer navigators) in improving the uptake of HIV testing, prevention and

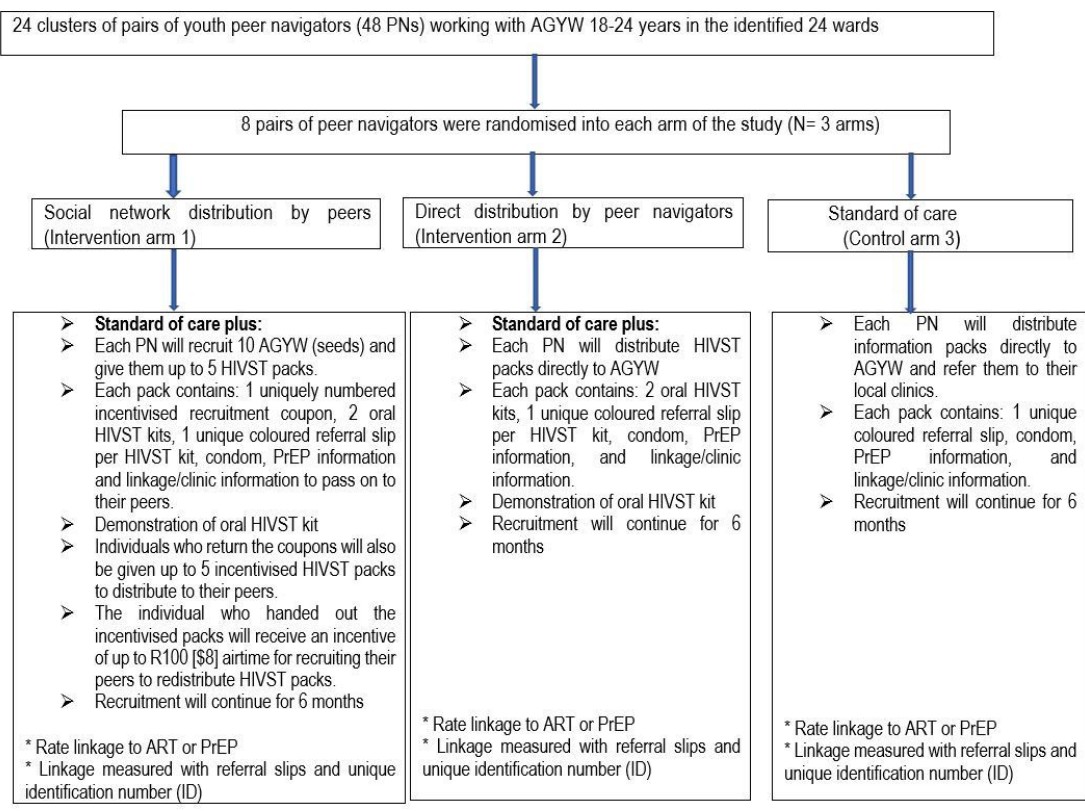

**Figure 2** Flow diagram of trial enrolment, randomisation and intervention arms. AGYW, adolescent girls and young women; ART, antiretroviral; HIVST, HIV self-testing; PNs, peer navigators; PrEP, pre-exposureprophylaxis.

care among young women (18 to 24 years). Eight pairs of peer navigators were randomised and assigned to each study arm with the intention of reaching young women aged 18 to 24 years with HIVST packs (including referral slips) and/or linkage information (including PrEP, contraceptives, ART etc) during the 6 month of community outreach. Peer navigators are randomised to one of three arms: 1) *incentivised-peer-networks:* peer-navigators recruited participants 'seeds' to distribute up to five HIVST packs (including incentivised coupons) and HIV prevention information to peers within social networks. Seeds receive an incentive (20 Rand = US$1.5) for each respondent who contacts a peer-navigator for additional HIVST packs to distribute; (2) *peer-navigator-distribution:* peer-navigators distribute HIVST packs and information directly to young people; (3) *standard of care:* peer-navigators distribute referral slips and information. All arms promote sexual health and HIV care and prevention (including PrEP and ART) and provide barcoded clinical referral slips to facilitate linkage to HIV testing, prevention and care services (figure 2).

The unit of randomisation is the pair of peer navigators working in each of the 24 areas included in the study. The areas are not adjoining, and each is bordered by a natural boundary (eg, roads or streams) or by a sizeable distance. Although contamination is inevitable in this type of cRCT, the spillover effects are contained by measuring the outcome by exposure to the peer-navigator cluster in multiple ways, including barcoded

and colour coded referral slips as well as peer-navigator and ward names that determine participant exposure to specific intervention components. Coupled with this, we are conducting a mixed method process evaluation that provides context and add nuance to our understanding of any contamination.

### Outcomes

The long-term goal of the intervention is to increase knowledge of HIV status and improve linkage to HIV care or prevention services such as PrEP among young women aged 18 to 24 years. A number of primary and secondary measures have been defined a priori. An interim analysis of the primary outcome will be conducted at 3 months.

### Primary outcome

The primary outcome compares the difference in linkage rate between arms, defined as the number of women (18 to 24 years) per peer-navigator month of outreach work (/pnm) who linked to clinic-based PrEP eligibility screening or started ART, based on HIV-status, within 90 days of referral.

### Secondary outcomes

The following calculations are planned for the secondary outcomes:

► Comparison of the difference per study arm of the total number of linkages (AGYW aged 18 to 24) per 100 clinic referral slips distributed.

► Comparison of the difference per study arm of the total number of linkages in men and women aged 18 to 30 per peer navigator outreach month.

► The change in proportion of young people aged 18 to 24 years who are aware of HIVST and who have used HIVST over time.

► Comparison of the difference per study area in the proportion of 18- to 24-year-olds who report knowledge of HIV status and uptake of ART, PrEP and VMMC in the surveillance area.

► The proportion of hard-to-reach adolescent girls and young women (aged 18 to 24 years) linked to care in the three study arms.

## Description of study arms

Intervention arm 1 - incentivised network distribution of HIVST: n=8 pairs of peer navigators are using RDS approach to distribute HIVST with health promotion and linkage information (eg, clinical referral slips and information about HIV and PrEP). Each pair of peer navigator recruits n=10, 18 to 24 years old female seeds from the participating communities. Each seed fills a brief *service recipient questionnaire* — self-filled on a tablet. Following which they receive verbal health promotion from the peer-navigator on the HIV prevention services available, the importance of sexual health, the benefits of HIV testing

PrEP and ART and a demonstration of HIVST. Seeds are asked to recruit AGYW aged 18 to 24 years preferentially but not exclusively and to avoid distribution of HIVST to those under the age of 18 or over the age of 30. All seeds are asked to complete a brief check of their understanding of the information provided to them, particularly information about not using HIVST if someone is on ART, the window period, the recommended support to those under 18 using HIVST, and the need for confirmatory testing.

As shown in figure 3, individuals who return with one of the coupons to a peer navigator undergo the same procedure as the seeds as described above. They are also given up to five uniquely numbered incentivised recruitment coupons and HIVST kits to pass on. When coupons are returned, the original individual who handed out the coupon receives a sum of R20 (US$1.5) in airtime per friend or peer who returns the coupon. This sum is a reimbursement for the time that they have spent in explaining and demonstrating the use of an HIVST and is not seen to be an undue incentive to coerce members of their social network to participate. There is no gender restriction of those recruited through the networks, however the primary outcome will be measured in young women aged 18 to 24 years only.

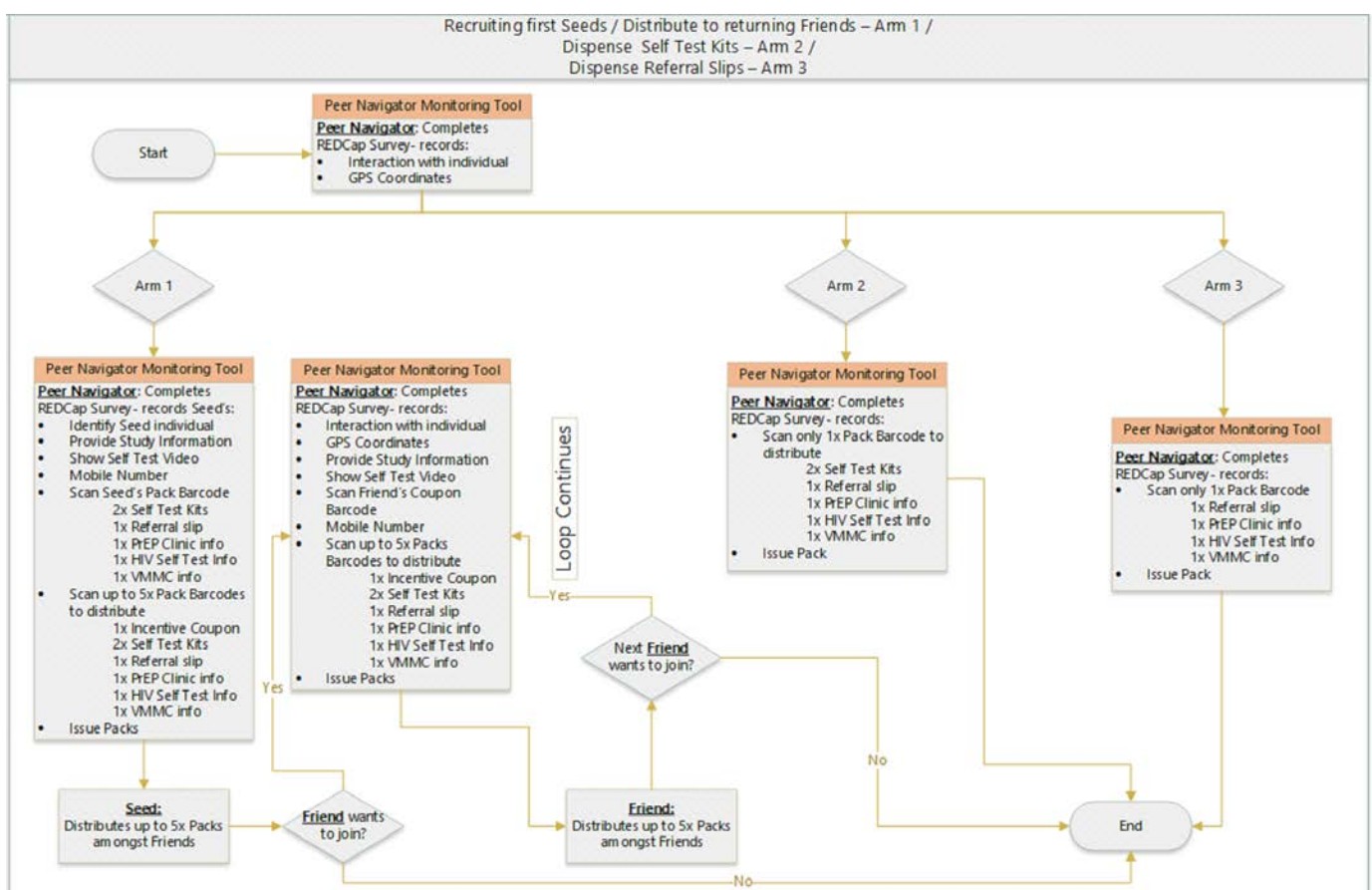

**Figure 3** Peer navigators community outreach workflow. GPS, Global Positioning System; PrEP, pre-exposure prophylaxis; REDCap, Research Electronic Data Capture; VMMC, voluntary medical male circumcision.

Intervention arm 2 - peer navigator direct distribution of HIVST: n=8 pairs of peer navigators are distributing HIVST packs with health promotion and linkage information (eg, clinical referral slips and information about HIV and PrEP) directly to young people aged 18 to 30 years over a period of 6 months. Each person contacted fills a brief *service recipient questionnaire* — self-filled on a tablet. Following which they receive verbal health promotion from the peer-navigator on the HIV prevention services available, the importance of sexual health, the benefits of HIV testing, PrEP and ART and a demonstration of HIV self-screening. All participants are asked to complete a brief check of their understanding of the information provided to them, particularly information about the unreliability of HIVST if someone is on ART, the window period, the recommended support to those under 18 using HIVST and the need for confirmatory testing.

Control arm 3: n=8 pairs of peer navigators are currently distributing packs with health promotion and linkage information (eg, clinical referral slips and information about HIV and PrEP) to encourage young people aged 18 to 30 years to test for HIV at clinics, and link to services/care. Each female aged 18 to 30 approached fills a brief service *recipient questionnaire* — self-filled on a tablet. Following which they receive verbal health promotion from the peer-navigator on the HIV prevention services available, the importance of sexual health, the benefits of HIV testing PrEP and ART.

## Study oversight

An independent scientific Technical Advisory Group (TAG) was formed by the Unitaid-funded HIV Self-Testing Africa initiative (STAR — a consortium of scientists conducting HIV self-testing related research in different countries) to monitor and supervise the progress of data collection, provide independent review of data collected during all cRCTs conducted under the STAR initiative, and assist investigators in disseminating results. TAG comprises members with expertise in HIV epidemiology, statistics, health economics, social science and AGYW. TAG will convene periodic meetings to review data and discuss any issues emanating from this trial.

## Study inclusion and exclusion criteria

Complete inclusion and exclusion criteria are summarised in tables 1–3. There are criteria for different recruitment stages in the trial. Above all, participants must not be less than 18 and not more than 30 years old. They must provide informed consent, not currently on ART and must be living in the study sites.

## Study recruitment and procedures

Study recruitment: Peer navigators are a cadre of recently matriculated youth or college graduates aged 18 to 30 years (male and female) recruited from the research community through the local municipal and traditional leaders. Between June 2018 to September 2018 participants underwent a 20 week training programme (3 days a

**Table 1** Inclusion and exclusion criteria for receiving the intervention, that is, the recruitment by peer navigators and/or seeds to receive HIVself-testing packs or clinical referral slips

| Inclusion criteria | Exclusion criteria |
| --- | --- |
| Participant must not be older than 30 years and younger than 18 years | Participants under 18 years or older than 30 years |
| Participant must agree to participate | Participant unwilling to participate |
| Both males and females can be included | None |
| Must not be known to be on ART – based on self-report | If on ART |

ART, antiretroviral.

week) which covered, youth development, HIV and sexual health information, HIV counselling and testing, confidentiality, ethics and research methods, study procedures and HIVST. Progress was evaluated using written and oral assessments to select 48 peer navigators to work in pairs and implement the intervention in their areas. The peer navigator intervention mirrors the South African cadre of community caregivers.

Before the peer navigators distribute the colour coordinated HIVST packs (each arm has its designated colour such as yellow, blue and pink) with unique identifiers in the intervention arms 1 and 2 or information packs in arm 3, the packs are being scanned and study participants are provided with information about the study and fill a brief *service recipient questionnaire* to be completed within Research Electronic Data Capture (REDCap - developed by Vanderbilt University, USA)[42] on a tablet (figure 3). Data collected at this point includes the date of recruitment, and the ID of peer navigator who recruited them, their age and area of residence. Participant's name, ID (eg, SA national number) and telephone or WhatsApp contact are optional. Those who are recruited through RDS will also be asked to provide data on their network

**Table 2** Inclusion and exclusion criteria for ascertaining the primary outcome

| Inclusion criteria | Exclusion criteria |
| --- | --- |
| Participant must not be older than 24 years and younger than 18 years | Participants under 18 years or older than 24 years |
| Provide written informed consent | Participants not willing to consent or unable to provide informed consent |
| Females | Males |
| Must not be known to be currently on ART | Currently on ART |

ART, antiretroviral.

**Table 3** Inclusion and exclusion criteria for ascertaining the secondary outcome

| Inclusion criteria | Exclusion criteria |
| --- | --- |
| Participant must not be older than 30 years and younger than 18 years | Participants under 18 years or older than 30 years |
| Provide written informed consent | Participants not willing to consent or unable to provide informed consent |
| Must not be known to be currently on ART | Currently on ART |

ART, antiretroviral.

size, barcode of the RDS coupons and the additional HIVST kits are scanned for further distribution.

Peer navigators spend ~30 min with each willing participant to explain the benefits of linking to care and prevention. Those explaining HIVST need 15 to 20 min extra and some young people require more time or more visits. This data is captured in REDCap for the purpose of process evaluation and costing. This data will be used in an aggregate way to understand the process and cost of the service delivery. Individualised data from the survey

will only be used in those participants that consent to their clinical data being linked and used for research purposes. If a participant withdraws their consent at any time, their data will be deleted from the research data set.

Study enrolment: Both walk-ins and study participants aged 18 to 30 years are eligible for receiving services from the designated study clinics. However, only those who have been referred through one of the three arms are eligible for study enrolment. This is being conducted by trained clinical research assistants in the designated study mobile and fixed clinics (figure 4). Also, all young women aged 18 to 24 years coming to one of the 11 primary healthcare clinics (PHC) or the mobile clinics in the surveillance sites are being directed by our AHRI data collection clerks to our research nurses. In both settings, the clinical research assistants or the research nurses explain the study and screen the young person for eligibility using a brief eligibility screening questionnaire on REDCap on a tablet. This includes questions to ascertain eligibility as well as arm of the study. If available, the clinical referral slip with the barcode with the unique identifier is scanned. The brief screening questionnaire that has further simple questions to ascertain if they were referred through any of the arms, that is, receiving any of the three colour coded packs/referral slips, or contact

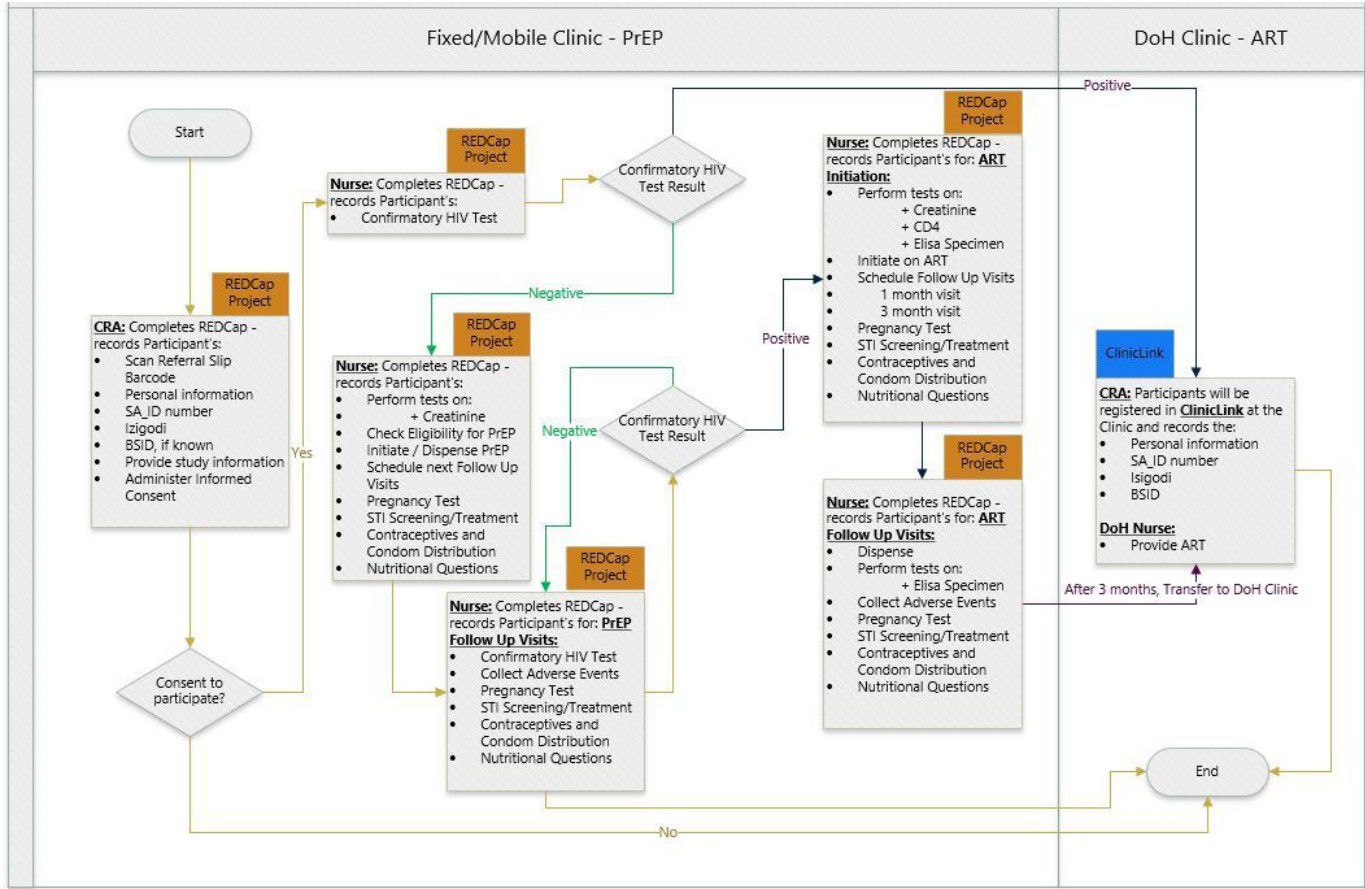

**Figure 4** Mobile/fixed clinics service workflow. ART, antiretroviral; BSID, Bayley Scales of Infant Development; DoH, Department of Health; PrEP, pre-exposure prophylaxis; REDCap, Research Electronic Data Capture; SA_ID,South Africa national identification number; STI, sexually transmittedinfection.

with a named peer navigator, or referral through peer network. Interested eligible participants then go through the process of informed consent. Participants provide written informed consent for enrolment into the study and specifically to use the information on their linkage to care as an outcome and to link their baseline questions.

Clinical procedures: Irrespective of whether they consent to the study, all eligible individuals attending clinics are offered confirmatory HIV testing, with two point of care tests and then blood sent to laboratories for ELISA testing (**Genscreen** ULTRA HIV Ag-Ab qualitative immunoassay (fourth Generation) Bio-Rad, Marnes-la-Coquette, France). Individuals who test HIV-negative receive counselling around the benefits of PrEP and HIV-positive individuals receive counselling around the benefits of immediate starting of ART. If they agree, they will undergo clinical screening for PrEP and ART. Screening includes Point of Care tests for creatinine (StatSensor-I create-test strips, Nova Biomedical UK) to assess renal function and hepatitis B (Alere Determine HBsAg, Alere International Limited, Ballybrit, Galway, Ireland; Architect i2000 analyzer, Abbott Diagnostics, Abbott Park, Illinois, USA), with vaccination offered to those who are negative, and sexual behaviour questionnaire to assess eligibility.

If HIV positive, ART is started by the professional nurses in any of 11 PHCs in the study sites. Patients who are eligible for PrEP are started in the three PrEP providing clinics (the mobile vans or the fixed urban adolescent and youth friendly clinic). Persons who are not eligible for PrEP receive counselling and, as indicated, a clinic referral with the screening results. A professional nurse initiates PrEP or ART usually on the same day or within 2 weeks of the screening visit. The professional nurse provides PrEP counselling that includes (1) *sexual health promotion,* with an emphasis on tackling the multiple health-related behaviours that will affect fertility and sexual pleasure (sexually transmitted infections, mental health, alcohol, diet and exercise); (2) assessment of fertility desire and contraception counselling; (3) choice of contraception and condoms; (4) HIV-negative men are also counselled around the benefits of VMMC and referred accordingly.

The counsellors provide counselling on adherence and develop an individualised adherence plan with the offer of face-to-face or virtual (WhatsApp/text based) adherence support. If the participant agrees to immediate PrEP initiation, s/he is issued with a month's supply of generic tenofovir disoproxil fumarate and emtricitabine. Baseline and follow-up bloods are taken and processed as per SA National Department of Health guidelines. The professional nurse registers the participant at the clinic (or updates the record if the participant is already registered) so that the participant's records are available should the participant seeks care there. Participants receive a phone call 7 days after initiating PrEP to complete a standard symptom screen for adverse effects and be referred to clinic for care if necessary. Participants have a clinic appointment scheduled 1 month, after PrEP initiation, as per national guidelines; appointments for refills and monitoring will be quarterly thereafter through either the mobile clinic, or other community-based refill points. Neutral text message reminders are provided for participants who have access to private messaging and phone calls. Participants are able to reschedule their appointments by text message, WhatsApp or calling the clinical hotline. Contact information is provided for the clinics whom participants can contact at any time.

## Randomisation
We defined cluster as a pair of peer navigator (PN) who live and work in one of the 24 administrative areas that were included in the trial. Peer navigators were preassigned to 24 areas before the randomisation process. Using data from our recent Determined, Resilient, Empowered, AIDS-free, Mentored and Safe (DREAMS, combination HIV prevention) Impact Evaluation study which collected data from a representative sample of young women residing in the study areas, a restricted randomisation was applied to get balanced covariates (location, HIV testing prevalence and uptake of DREAMS combination HIV prevention by adolescent girls and young women) across the three arms. We generated a random set of possible 100 000 allocation options. After applying all three restrictions, a total of 47 924 possible allocations remained, and a random number was generated and assigned to each allocation option. The random numbers were ranked from lowest to highest and the allocation option with the first rank was then selected. A randomisation list with intervention arms named alphabetically (A, B and C) was generated.

Following the statistical randomisation, a public randomisation was conducted where peer navigators were divided into three groups (A, B and C). Each group had 16 allocated PNs and three floating ones. Each group chose a suitable name and a leader who represented them. Group leads picked a concealed number to determine the order of picking their study arm from a box. The facilitator shook the box so to make sure that each concealed arm in the box had an equal chance of being picked. Lastly, the leaders were asked to open and announce the arms of their respective groups to the bigger group.

## Blinding
The statistician and clinical staff did not participate in the public randomisation with peer navigators and they will remain blinded until the results of the study have been finalised.

## Sample size calculations
Based on 2017 data, we estimated ~500 age eligible 18 to 30 year olds live in each peer navigator team catchment area, of whom we anticipated at least 200, 18- to 24-year-old females will be handed a study pack (so cluster size at least 200). We estimated this based on two peer navigators working approximately 1000 hours over the study

period per cluster. We estimated that they would reach two young adults per each 4 hour of work and at least one would accept a study pack. We calculated the sample size calculation using the primary outcome, the rate of linkage after 90 days among women ages 18 to 24 years. Using our existing data on uptake of HIV testing in the DREAMS interventions as well as our data on uptake of testing and linkage to HIV care in the demographic surveillance rounds of, we estimated that one woman would link per 6 months of peer navigators outreach work time in the standard of care. With eight peer educator pairs (or clusters) per arm and a cluster coefficient of variation (k) of 0.25, we will have 80% power to detect a 100% increase in rate from 1 woman to 2 women per 6 months of follow-up, and 90% power to detect a 150% increase from 1 woman to 2.5 women per 6 months of follow-up. We have chosen policy and clinically relevant increases in linkage to care. Assuming additional clustering of the outcome within peer educators and increasing the coefficient of variation (k) to 0.35, we have 80% power to detect a 150% increase in rate from 1 woman to 2.5 women per 6 months of follow-up. All sample size calculations assume two-tailed statistical tests with alpha=0.05.

## Process evaluation

Our aim is to assess the acceptability, feasibility and fidelity of the peer delivery model in each arm in facilitating linkage to care. We compare the pattern of recruitment per arm and assess the proportion of hard-to-reach AGYW (aged 18 to 24 years) — defined as out of school, recently migrated and those who live in remote areas linked to care in the three study arms. We will also explore potential unintended consequences and ethical issues that arise during peer referral and HIVST and ascertain what works for whom and when to be able to modify the intervention to improve equitable reach and coverage. Specifically, we will explore the reach of network recruitment compared with peer outreach work, in terms of reaching more vulnerable groups (out of school, recently migrated and those who live more remotely). Entrenched in realist evaluation, this process evaluation uses a mixed method approach (quantitative and qualitative research techniques) to investigate implementation, mechanisms of impact and contextual factors, informed by the UK medical research council guide[43] and wider implementation science literature with a focus on fidelity, reach and acceptability.[44]

## Cost-effectiveness evaluation

We compare the costs in intervention and control arms. Cost per case linked to PrEP eligibility assessment (HIV-negative) and cost per case started on ART (HIV-positive). To establish costs, we are using both a bottom-up ingredient-based costing approach and a top-down costing approach using the study budgets and expenditure reports. Specifically, we calculate and cost the actual time spent by peer-navigators in each arm for each person linked to care and prevention.

## Data collection
### Participant survey and clinic linkage
A short survey (service recipient questionnaire) is administered to consenting individuals participating in the study. The questionnaire collects data on participant's demographic information and coupon identification. The data is captured on REDCap on a tablet. The survey takes approximately 5 min to complete and is administered in both English and isiZulu. The primary outcome of linkage is measured through identifying the consenting eligible young women who link to care through the 11 PHCs and the mobile clinics. We use an algorithm to identify which arm the individual came from, including the barcode on the referral slip they bring, the colour of the referral slips or HIVST pack, their area of residence and the identity of the peer-navigator that recruited them.

### Programmatic data
In addition to the survey, we collect the programme data records from the peer navigators daily reporting of their outreach activities. This includes the number of young people they have counselled and the numbers they have referred to services and the brief service recipient data they have collected on those who have received referral slips or HIVST packs. We use the programme data in an aggregate way (disaggregated only by gender) to understand the reach and coverage of the programme and compare that with those who link to care. We also use data on changes in self-reported HIVST and linkage services collected through the population intervention surveillance platform.

### Participant in-depth interviews
In-depth interviews (IDIs) are being conducted with the peer navigators (n=30), clinical team (n=6) stationed in clinics in the participating communities and a purposive sample of young women aged 18 to 24 years (including young men n=45; 30 females (10 per arm), 15 males (five per arm)) across the three arms and clinics. The interviews are conducted by trained social scientists fluent in English and isiZulu and take approximately 60 min in length depending on the participant's responses, and this enables the researchers to understand, contextualise and explore participants perceptions of the study and some of the issues emanating from the trial. The small number of IDI participants in qualitative study is allowed since deeper meanings of concepts and thematic areas are explored. To limit disturbances and ensure privacy, the IDIs are conducted in a private space suitable for the participant, and audio recorded with interviewees' consents. Prior to the interview, participants are encouraged to use pseudo names instead of their real names.

### Statistical analysis
The analysis of primary outcome follows an intention-to-treat (ITT) and per-protocol approaches. The primary outcome compares the difference between the rate of linkage of 18- to 24-year-old women to HIV confirmatory

HIV testing, ART (if HIV-positive) or PrEP counselling (if HIV-negative). The rate is defined as the number of linkages per month of peer navigator outreach activities. The numerator is defined as the number of young women aged 18 to 24 who attend clinic for confirmatory HIV testing, PrEP counselling or ART, following HIVST distribution or peer navigator referral to HIV testing, treatment and prevention services. The denominator for ITT analysis is the entire time (study duration) spent by peer navigators doing their peer outreach work. For the on-treatment analysis, we will use the actual time spent by peer navigators on distributing packs in each arm. The time worked by each peer navigator will be combined to get the total time per pair of peer navigator. The difference in rate of linkage between the study arms will be calculated — incentivised HIVST delivery through peer network and direct distribution of HIVST will be compared with standard of care.

Since we randomised the pairs of peer navigator (clusters), the rate of linkage will be calculated for each pair of peer navigator using aggregate data for each cluster. Since the number of clusters are small, the effect of the intervention will be estimated using a two-stage approach based on cluster-level summaries.[45] The cluster-level approach, although less statistically efficient than methods based on individual level regression, is more robust when there are a relatively small number of clusters. All analyses will be performed using Stata V.15 (StataCorp LP, College Station, Texas, USA).

Cluster-level linkage rates will be calculated and used to estimate the unadjusted rate ratio and its 95% CI for the effect of each intervention arm compared with the standard of care; the mean difference in linkage rates between each arm and standard of care, and against eachother will be assessed using a t-test. A rate ratio adjusting for substantial covariate imbalance at baseline will also be calculated, using a two-stage process; all covariates will be prespecified in the analysis plan. To identify covariates for adjustment, baseline characteristics of each arm will be presented, and the size of the difference of covariates known to be associated with the outcome will be assessed quantitatively.

As part of the exploratory analysis, we will perform a (i) subgroup analysis by gender and area and (ii) two intervention arms will be compared with one another (incentivised HIVST delivery through peer network approach will also be compared with direct distribution of HIVST approach). To expand on this, the data from the client survey captured on REDCap dashboard will be exported into Stata, cleaned and analysed. All reporting will conform to Consolidated Standards of Reporting Trials guidance for cluster randomised trials.[44 45]

## Qualitative analysis

NVivo software will be used for categorisation and coding of emerging themes from the interview transcripts. Identified themes (including participants' quotes) and interview transcripts will be reviewed and compared by the research team for inconsistencies and adequate representation of participants' views. Emerging themes that address the key focus of the study will be examined and analysed following an interpretivist approach.[46]

## Adverse events reporting

HIV testing, including HIVST, is well established and known to have a high level of safety. However, harmful reactions can occur. Adverse events (AE) related to HIVST include all undesirable experiences that result directly from use of the HIVST kit itself or as a reaction from others due to the presence of the kit, use of the kit or results produced from the kit. AEs can be from one person to another, or a person to themselves, and can occur before, during or after self-testing. We rely on participants to report any AEs to the study staff or through the hotline provided on the referral slip. Also, during PrEP resupply and monitoring visits, participants complete a standardised symptom screening questionnaire for adverse effects of PrEP as per South African clinical guidelines. Furthermore, all participants will receive regular creatinine tests to monitor their renal function. Participants who have severe (grade 3/4) adverse effects and serious adverse effects, are referred to the study clinician for medical evaluation. All participants who experience adverse events receive follow-up until the adverse event is resolved.

AEs and serious adverse events (SAEs) are captured through the process evaluation and community engagement units and the telephone hotline. In addition, peer navigators and clinic staff log AEs using our incident reporting form for up to 12 months after the start of the intervention. Reported AEs and SAEs are monitored, categorised based on an established grading system. SAEs are logged, with the Principal Investigator to evaluate the SAE for seriousness and likely relationship to the intervention. Related SAEs are reported to University of KwaZulu-Natal (UKZN) and London School of Hygiene and Tropical Medicine (LSHTM) Ethics Review Boards. All SAEs are reported regularly through 6 month progress reports to TAG members, local and international collaborators. Annual reports with full listings of SAEs will be submitted to Ethics Review Boards.

## Data management

Quantitative data are collected directly on the study tablet via REDCap database and resides within a single MySQL database server within a secure server cluster. Study-specific electronic laboratory results are transferred directly to a secured server for storage. Qualitative data are stored in the form of Word files or in Excel both of which can be uploaded into NVivo qualitative data management programme. The use of MS Word will ensure that data can in future be shared for use in different analysis programmes. These files will be kept on a secure access-controlled folder on a secured server. Qualitative audio files will be destroyed once they have been transcribed, translated and quality controlled.

## Patient and public involvement

Although we did not involve patients or the public in the design of the study, findings from previous studies conducted within the community were useful during the study design phase. The study was also presented to the community advisory board and the district department of health for comments before it was submitted to Institutional Review Boards for ethics approval. The results of the study will be shared with the peer navigators and the research community through community dialogues and the community advisory board.

## ETHICS AND DISSEMINATION
### Ethical consideration: confidentiality and informed consent

All staff (including peer navigators) have been provided with training on research ethics such as confidentiality, voluntary participation and good clinical practices. Anonymity and confidentiality are ensured at all levels of the research process, and none of our reports, presentations or articles will contain study participants identifying information. Pseudonyms are used when reporting the data particularly qualitative data. Each participant is assigned a unique non-identifying participant identification number. Prior to their involvement in the study, participants are provided with adequate information about the study and are allowed to ask questions for clarifications. Voluntary informed consent is collected only if participants have the full understanding of the study procedures. A copy of the signed consent form is given to them. Participants are informed about the importance of the confirmatory diagnostic testing. An anonymous support hotline is provided on the referral slips should they need to discuss their HIV status, counselling and further linkage to HIV care or other health services. All participant irrespective of their consent to participating in the study are eligible for the clinical services provided through the study. The study was approved by and conforms to the ethical guidelines and standards of UKZN, LSHTM and WHO.

### Dissemination plan

The results of this study will be disseminated through traditional academic channels (peer-reviewed journal publications) as well as on different information dissemination platforms such as conferences, workshops, community meetings and symposia. The results of the study will also be presented at Self-Testing Africa (STAR) consortium meetings and will be included in the WHO guidelines. The results of the study will be shared with the peer navigators and the research community through community dialogues and the AHRI community advisory board. A detailed findings report will be shared with the Department of Health and other stakeholders to inform policy.

## DISCUSSION

Despite the burden of HIV and the availability of free HIV testing and treatment in our local PHCs, HIV status knowledge remains low among young people <30 years[2]. Several complex barriers (eg, stigma, confidentiality, family rejection, waiting times and lack of youth friendly services etc) impeding on young people's access to HIV care services were identified in a formative research we completed in 2018.[2 38] Studies including systematic reviews[13 19] have shown HIVST to be a promising alternative HIV testing option because it is private, flexible and an efficient method. However, there is limited evidence on the use of HIVST to improve linkage to prevention such as PrEP. The overarching goal of this trial is to address the gap in HIV testing and linkage to HIV prevention among young women aged 18 to 24 years. This goal will be achieved through the assessment of two HIVST delivery models (incentivised peer network vs direct distribution by peer navigators) compared with the standard of care of peer navigator only.

A major strength of this study is the development of a theoretically derived intervention that can be implemented through existing cadre of community caregivers and peer-to-peer networks across sub-Saharan Africa.[47–50] If found to be effective in increasing HIV testing uptake and prevention, the intervention is designed to be rolled out. Also, using a rigorous cost-effectiveness analysis will allow South African policymakers to evaluate the cost-benefit ratio of using the different models of distribution in different settings. Furthermore, by collecting rigorous data on linkage to prevention both through the trial and from our surveillance infrastructure, we can understand the potential population impact of the different methods of HIVST distribution on knowledge of status and linkage to care and prevention. Ultimately this will provide evidence of the potential of the intervention to attract young people into the HIV care and prevention cascade and inform the evidence base to reduce the mortality and morbidity in youth. Lastly, rigorous process evaluation and collection of data on all adverse events and social harms will provide important data around some of the concerns about HIVST, that is, the potential for coercive test, depression, anxiety, suicidality or intimate partner violence.

In order to reduce the risk of contamination due to the proximity of the areas and the nature of the intervention, we have piloted several methods including the use of referral slips with unique codes and colour coordinated packs to identify the arm individuals come from when they link to care. Previous data from nested cohorts that we have followed up in our area has shown that young people rarely migrate within our surveillance area given that only 220 of 2184 young people <25 years cohorts moved from one cluster to another in 2017/2018. Furthermore, the process evaluation will help us understand the delivery model in each arm as well as unintended consequences and ethical issues that arise during the study and ascertain what works for whom and when to be able to modify the intervention to improve equitable reach and coverage.

In conclusion, the results of this trial are expected to contribute to WHO guidelines and informed policy aimed at implementation and scale-up of HIVST and PrEP in South Africa. Also, this study will address critical gaps in the literature on HIV testing and prevention interventions for young people particularly females aged 18 to 24 years in Southern Africa.

**Author affiliations**
[1]Africa Health Research Institute, Mtubatuba, KwaZulu-Natal, South Africa
[2]Sociology, University of Johannesburg, Auckland Park, Gauteng, South Africa
[3]Epidemiology and Public Health, London School of Hygiene and Tropical Medicine, London, UK
[4]Department of HIV/AIDS, World Health Organization, Geneva, Switzerland
[5]Population Services International, Harare, Zimbabwe
[6]Public Health and Policy, London School of Hygiene and Tropical Medicine, London, United Kingdom
[7]International Public Health, Liverpool School of Tropical Medicine, Liverpool, UK
[8]CeSHHAR Zimbabwe, Harare, Zimbabwe
[9]Infectious and Tropical Diseases, London School of Hygiene and Tropical Medicine, London, UK
[10]Institute for Global Health, University College London, London, UK

**Acknowledgements** The authors acknowledge TAG of the STAR initiative and AHRI HIV Prevention Multilevel Group including the research assistants, peer navigators, clinical team, managers and research administrators for their commitment to the study. We also extend our appreciation to our research community including the community advisory boards in uMkhanyakude district.

**Contributors** MS and LC conceived the study. OA, MS, CH, JD, JS, LC, FC, NC, NO, PM, NM and MN designed the study. OA and MS wrote the first draft of the manuscript. OA, MS, LC, JS, JD, CH, TS, FC, MN, NC, KH, CJ, PM, NM and NO read and critically revised the manuscript. All authors read and approved the final manuscript.

**Funding** This study is part of the Self-Testing Africa (STAR) initiative funded by the Unitaid (grant number: PO#10140-0-600). MS is partly supported by the US National Institute of Health (NIH) R01 (award no: 5R01MH114560-03) that supports a peer led outreach team of navigators to support uptake and retention of adolescents and young adults in existing HIV prevention. Africa Health Research Institute is supported by core funding from the Wellcome Trust (Core grant number (082384/Z/07/Z)).

**ORCID iD**
Oluwafemi Atanda Adeagbo http://orcid.org/0000-0003-1462-9275

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
