## [Reviewer comments · BMJ Open]

ARTICLE DETAILS

TITLE (PROVISIONAL)	A cluster randomised controlled trial to determine the effect of peer delivery HIV Self-Testing to support linkage to HIV prevention among young women in rural KwaZulu-Natal, South Africa: study protocol
AUTHORS	Adeagbo, Oluwafemi; Mthiyane, Nondumiso; Herbst, Carina; Mee, Paul; Neuman, Melissa; Dreyer, Jaco; Chimbindi, Natsayi; Smit, Theresa; Okesola, Nonhlanhla; Johnson, Cheryl; Hatzold, Karin; Seeley, Janet; Cowan, F; Corbett, Liz; Shahmanesh, Maryam

VERSION 1 – REVIEW

REVIEWER	Lucia Knight School of Public Health, University of the Western Cape
REVIEW RETURNED	30-Aug-2019

GENERAL COMMENTS	General comments: Thank you for the opportunity to review an interesting paper and a very relevant protocol. The challenges of presenting large amounts of complex information in a relatively small space is acknowledged but some significant revisions are required to make the processes clearer and allow for the presentation of the protocol to be more coherent and sensible. In particular, the methods section could benefit from some reordering of sections to improve the flow and clarity of the proposal. In addition, possibly reconsider the sub-section headers to aid this given the guidelines are flexible (consolidate and reduce). If this is undertaken it is possible that some of the questions/issues raised in the review, especially in the first few sub-sections, could be ignored because they will be addressed by better organisation. This paper requires thorough proof reading and editing for language and punctuation errors before it should be considered for publication. Introduction: • The introduction makes mention of young men in reporting the results of the TasP study and yet the focus here is on young women- some greater clarity and justification for this would be useful.• HIV care services should be defined in the first paragraph• Remove the insert starting “-which refers to” because it is much better described below• Comment on acceptability and uptake of HIVST in SA specifically• Unnecessary repeat of accuracy in second paragraph• “HIVST kit product” is unclear and example of grammatical errors within the paper• The third paragraph fails to clearly distinguish between PrEP for prevention and ART for treatment and doesn't provide adequate
--

	justification for focus on both separately  • Oral-based HIVST should be revised for more standard language • The inclusion of only older adolescents and cut off of 24 years would be useful to justify • “higher risk” than who? Specific aims  • This aims are fairly clear but the justification for the differences between each are not. For example the study intro etc. focus on young women aged 18-24 and yet this is not the focus of aims 1 and 3. • Greater background for and justification of these differences is required. METHODS Theory of change  • It is not clear how the mental models and deductive development have influenced the theory of change or what exactly the theory of change being employed in the study clearly is. • Greater linkage to the theory and models and clear evidence for how these influence the concepts being employed in the study is required. • In addition, the link between the formative work and theory being proposed for use here is not clear especially the second citation of formative work. Trial design  • This section is quite repetitive in itself with multiple reference to each arm and what it entails. • The number of arms being referred to shifts in the first paragraph with three, two and then three again- ensure consistency. • It is not clear how the target of 5000 young women has been reached. • The target per arm does not add up to the total number of women to be targeted by the intervention. • The process of randomisation of community and peer navigators is not adequately described. • How were communities selected and what is the geographic distribution of these how were community level differences considered? Description of the Study arms  • This section is very repetitive of the preceding section. • Arm 1  o How will the seeds be recruited? Are there any other criteria other than age? o What sort of session will be undertaken? How long will it take? How will peer recruits be prepared for this? o The age limits for recruitment of AGYW requires some explanation. o Clarify the meaning of this “particularly the unreliability of HIVST on ART” o Coupon return is introduced with not previous mention/context o Specify “the same procedures as seeds” o It is not clear what seeds are being expected to do • Arm 2  o Does recruitment= distribution of a pack? o What sort of session will be undertaken? How long will it take? How will peer recruits be prepared for this? o The age limits for recruitment of AGYW requires some explanation. o Clarify the meaning of this “particularly the unreliability of HIVST on ART” • Arm 3
--	--

	o What does a linkage information pack include? o “encourage” should be “encouraging” o What sort of session will be undertaken? How long will it take? How will peer recruits be prepared for this?  • Many of these things could be defined and explained just once to provide greater clarity. Study oversight  • “convey” should be “convene” • What was the role of the TAG in the study design? Study recruitment and procedures  • “fresh graduates” is not clear • Provide some more details about recruitment of a training of peer navigators, what is the “several training and assessments” referred to? • 5000 young women overall? Ensure this is clearly stated throughout. • How were the NDoH involved in the design of the intervention? • See above with regards to how the estimated targets were calculated? • What do the packs include and how are they colour coordinated and for what purpose? • “Scanned in”? • Who are the participants being referred to in the first paragraph? • “The information is relevant”- what is the information that this is referring to, data collected in the questionnaire? • How is the collection of process evaluation data collected in a different way from the data collected from participants? • How are the participants in paragraph 1 different from those in paragraph 2. • Will individuals enrolled in the facilities be enrolled regardless of interaction with peer recruiters in any arm? The eligibility criteria are not clear. • Do those enrolled at facilities complete any form of data collection process more than the screening? • The start of paragraph 3 fails to make clear whether this will be all visitors to the clinic or eligible visitors? • If all clinics don’t all offer ART and PrEP how will referrals be made and information about uptake be linked? • Possibly split this section (final two paragraphs) to deal with HIV+ and HIV- separately to make sure processes are more clearly described. Inclusion and exclusion criteria  • May be more sensible to incorporate in recruitment and doing this by phase for greater clarity. • There is limited information about or justification for required/targeted sample size for the study. • The primary and secondary outcome as noted in the tables are not completely clear- only defined two pages after this section. Randomisation  • What is the PIPSA? • What were the allocation restrictions? • How do allocations map onto areas? • What is a PN? • This section requires reworking for clarity- it is quite confusing. Sample size  • This needs to be discussed much earlier to help with clarity of process and flow. • What is the estimated population size per area? Outcomes  • The specific outcomes should be possibly discussed and
--	--

	presented alongside the study aims, the tools and analyses are better placed in an analysis section.  • What is the justification for the interim analysis? This should be placed within data collection. • The primary outcome should be rewritten for more clarity. Difference with what?  • Formatting changes in this section. Process evaluation  • If possible include all methods relating to this in one section. • Why will patterns of recruitment be considered in the evaluation? • How will unintended consequences and ethical issues be assessed? • How will reach will assessed? • The use of and application of the MRC guidelines are not clear. Data collection  • Is the voucher the same as the coupon? • Is the survey the same as the service recipient questionnaire? • How will the sample for IDIs be selected and based on what criteria? • What will issues/topics will the IDIs cover? Adverse events  • Seems logical for this to be addressed along with other ethical issues. • How will any instances of AE relating to HIVST be picked up by the study given these are largely circulated and used within the community? • Community engagement units and the telephone hotline are not described nor explained/given context. Analysis  • Link more clearly with outcomes • Describe the qualitative analysis using an accepted method. Discussion  • The argument about the intervention being derived from theory is not supported by the theory presented in the paper. • The discussion presents a number of AEs, social harms and potential negative outcomes not addressed in the AE or ethics section- it is unclear how the study will be able to collect data on these. • Process issues including referrals and colour coding are not adequately discussed within the methods.
--	--

REVIEWER	Adrienne Shapiro University of Washington, United States
REVIEW RETURNED	14-Sep-2019

GENERAL COMMENTS	The authors have written a clear, well-motivated protocol for a 3-arm, randomized controlled trial of interventions to increase uptake of HIV testing and linkage to prevention & care through use of peer-delivery HIV self-test kits and education. The protocol clearly outlines the study design and rationale, study methods for recruitment, enrollment, and outcomes to be measured and compared. Importantly, the authors also include qualitative methodology to further investigate the outcomes beyond numeric percentages, as well as cost-effectiveness studies to assist with translating protocol outcomes into policy recommendations.
--

REVIEWER	Handan Wand The Kirby Institute, UNSW, Sydney Australia
REVIEW RETURNED	22-Oct-2019

GENERAL COMMENTS	This is a very well written manuscript. I have minor comments: I am not sure why there were two inclusion/exclusion criteria one for i.e. the recruitment by Peer Navigators and/or Seeds to receive HIVST packs or clinical referral slips which was specified as 18-30 years; while, primary and the secondary outcomes will be ascertained among 18 to 24 years old Page 12: is there any reason that the lines 9-10 used bold/bigger format Statistical analysis: It is not clear how the primary and secondary outcomes will be analysed? Investigators stated that they will be using t-test and 95% CIs. What will be compared using the t-test? Cluster-level summaries? On Page 14: what does "standard methods" means? Investigators do not plan any adjusted analysis; however, it is usually a custom to suggest an adjusted analysis in case there was a significant imbalance between the study arms. Is there a reason that the investigators do not expect any imbalance? Then, later in statistical analysis section (page 14 lines 24-27), it was stated that "Substantial differences will be identified by comparing frequencies or means of variables known to be associated with the primary outcome. These will be assessed by investigators without the use of statistical tests." My question is: How the substantial differences will be assessed without the use of statistical tests? Do they mean "no p-values will be presented"
--

VERSION 1 – AUTHOR RESPONSE

Reviewer 1:

General comments

=====

Thank you for the opportunity to review an interesting paper and a very relevant protocol.

The challenges of presenting large amounts of complex information in a relatively small space is acknowledged but some significant revisions are required to make the processes clearer and allow for the presentation of the protocol to be more coherent and sensible. In particular, the methods section could benefit from some reordering of sections to improve the flow and clarity of the proposal. In addition, possibly reconsider the sub-section headers to aid this given the guidelines are flexible (consolidate and reduce). If this is undertaken it is possible that some of the questions/issues raised in the review, especially in the first few sub-sections, could be ignored because they will be addressed by better organisation.

Authors' Responses to Reviewer 1:

Thank you very much for your constructive comments. They are well received and have been

attended to. Please see below:

Introduction:

- The introduction makes mention of young men in reporting the results of the TasP study and yet the focus here is on young women- some greater clarity and justification for this would be useful.

RESPONSE: Paragraph one of the background has been revised as suggested to emphasise the focus is on women. Thank you.

- HIV care services should be defined in the first paragraph

RESPONSE: This has been defined in the manuscript as suggested as “HIV testing and uptake and adherence to antiretroviral therapy for treatment”.

- Remove the insert starting “–which refers to” because it is much better described below

RESPONSE: Deleted as suggested.

- Comment on acceptability and uptake of HIVST in SA specifically

RESPONSE: Thank you. Paragraph 2 has been revised and additional SA literature was added:

- Knight L, Makusha T, Lim J, Peck R, Taegtmeier M, van Rooyen H. “I think it is right”: a qualitative exploration of the acceptability and desired future use of oral swab and finger-prick HIV self-tests by lay users in KwaZulu-Natal, South Africa. BMC research notes. 2017;10(1):486.

- Makusha T, Knight L, Taegtmeier M, et al. HIV self-testing could “revolutionize testing in South Africa, but it has got to be done properly”: perceptions of key stakeholders. PloS one. 2015;10(3):e0122783

- Unnecessary repeat of accuracy in second paragraph

RESPONSE: Deleted. Thank you.

- “HIVST kit product” is unclear and example of grammatical errors within the paper

RESPONSE: This has been edited in the manuscript as suggested.

- The third paragraph fails to clearly distinguish between PrEP for prevention and ART for treatment and doesn't provide adequate justification for focus on both separately

RESPONSE: Thank you. We have stated the existing evidence for the effectiveness of community-delivery/ peer support and HIVST in linkage to HIV care. We have then clarified that there is a gap in the evidence-base for the use of HIVST to link to prevention. This paragraph has been added, “Moreover, there is some evidence to suggest that HIVST can improve linkage to treatment when coupled with community based support. However, there is limited evidence of the effectiveness of HIVST to link people who are negative to prevention, and in particular PrEP, with or without community-based support.”

- Oral-based HIVST should be revised for more standard language

RESPONSE: This has been edited accordingly.

- The inclusion of only older adolescents and cut off of 24 years would be useful to justify

RESPONSE: Adolescent girls and young women (aged 15-24) have been defined by the South African NDoH as a key population for targeting HIV prevention, including PrEP, due to ongoing high HIV incidence. National guidance has suggested that HIVST in those under age 18 years should be supervised by a health care worker and for this reason, although the community promotion of HIV testing and linkage to care was provided to men and women aged 18-30 years, we have focused on measuring the primary outcome of linkage to PrEP in those women aged 18-24 years.

We have reordered the background to ensure that the persistent challenge of high incidence of HIV in young women aged 15-24 is clear (paragraph 1); explain the NDoH guidance in HCW supported testing in those <18 (para 3) and NDoH guidelines on PrEP targeting (para 4); and then in last paragraph of the background we have clarified why the intervention is delivered to young people aged 18-30 years but the outcome of linkage to PrEP is measured in those aged 18-24 years.

- “higher risk” than who?

RESPONSE: Higher risk has been deleted accordingly.

Specific aims

- These aims are fairly clear but the justification for the differences between each are not. For example, the study intro etc. focus on young women aged 18-24 and yet this is not the focus of aims 1 and 3. Greater background for and justification of these differences is required.

RESPONSE: Thank you very much for this observation. We have justified this in the introduction accordingly. Although males and females aged 18-30 years are included in the study our primary outcome will be determined in AGYW 18-24 years (see the explanation above).

METHODS

Theory of change

- It is not clear how the mental models and deductive development have influenced the theory of change or what exactly the theory of change being employed in the study clearly is. Greater linkage to the theory and models and clear evidence for how these influence the concepts being employed in the study is required.

RESPONSE: Thanks for your observation. This has been revised accordingly. Both mental models and deductive development are embedded in ecological approach which is the main framework of the trial. Our intention is to explain or test the causal relationships between concepts and the multiple factors (e.g. individual beliefs, family, community, social networks, cultural norms, health systems etc.) intersections that shape young people’s access to HIV testing, treatment and prevention. The theoretical stance will be unpacked when reporting the findings of the study. Thus, our overarching theory of change is inherent in the ‘ecological approach’ and will be explored in the process evaluation of the study.

- In addition, the link between the formative work and theory being proposed for use here is not clear especially the second citation of formative work.

RESPONSE: Thanks for your observation. This has been teased out in the manuscript.

Trial design

- This section is quite repetitive in itself with multiple reference to each arm and what it entails.

RESPONSE: This has been edited accordingly.

- The number of arms being referred to shifts in the first paragraph with three, two and then three again- ensure consistency.

RESPONSE: Thank you for highlighting this. The inconsistency has been dealt with in the manuscript.

- It is not clear how the target of 5000 young women has been reached. The target per arm does not add up to the total number of women to be targeted by the intervention.

RESPONSE: This is a cluster Randomised Controlled Trial nested in a demographic surveillance area where we have triannual household visits that enumerate the population living in the area. Based on the 2017 demographic surveillance data, we know that ~12000 men and women aged 18-30 years are living in the area of which the target population is young women aged 18-24 years. Our sample size calculation was based on this target of ~200 women aged 18-24 years could be reached per cluster. We have clarified the language to explain what we mean by the different populations.

- The process of randomisation of community and peer navigators is not adequately described. How were communities selected and what is the geographic distribution of these how were community level differences considered?

RESPONSE: We have rewritten the study setting and study population and placed it earlier in the manuscript to make the randomization when it is described under 'randomization' clearer as described below:

"Study setting and population:

This study will be conducted in Africa Health Research Institute's (AHRI) long-standing demographic surveillance area in northern KwaZulu-Natal. The study area is mostly rural and poor compared with other parts of South Africa, with high levels of unemployment (over 85% of young adults aged 20-24 are unemployed) and the local language is IsiZulu. In the study area 8 out of 100 women aged between 20 and 24 acquire HIV in one year and 4 out of 10 women attending antenatal clinics are found to be infected with HIV. Data between 2011 and 2015 in the study area suggests that sexually active women aged 16-29 and young adult men have an HIV incidence above the threshold of eligibility for PrEP.

The demographic surveillance area provides over 16 years of household history, and over a million person-years of follow-up through annual individual-level surveys, which capture sexual behaviour and partnerships, reproductive histories and contraception use, access to HIV testing and care, access to HIV prevention services (including VMMC), as well as socio-demographic information.. Moreover, through a Memorandum of Understanding with the Department of Health AHRI has also embedded data collection clerks within the public health clinics to capture electronically any clinical attendance and linking it with the surveillance platform on all consenting attendees. This allows us to measure linkage of individuals to HIV care and use of contraceptive services.

As part of a NIH R01, we have selected, trained and employed 24 pairs of peer navigators, working in 24 discrete areas (based on administrative divisions) of the demographic surveillance area (the Hlabisa district of uMkhanyakude district of northern KwaZulu-Natal (KZN), South Africa) to deliver HIV and sexual health related Health promotion to an estimated 12000 male and female youth aged 18-30 years (~500 per each of the 24 administrative areas) and young women aged 18-24 years residing in the 24 areas (figure 2).”

Description of the Study arms

- This section is very repetitive of the preceding section.

RESPONSE: This is noted, and we have attempted to cut down on repetitions.

Arm 1

- How will the seeds be recruited? Are there any other criteria other than age?

RESPONSE: The seeds are recruited by peer navigators working in the study sites. Age and willingness to distribute the HIVST packs to their friends are key criteria. They must also be from study sites.

- What sort of session will be undertaken? How long will it take? How will peer recruits be prepared for this?

RESPONSE: Thank you for noting that this was not clear. We have added this clarification,

“This is a brief verbal information where the peer navigators provide potential participants with information about health and social services available in their areas. Between 6/2018-9/2018 participants underwent a 20-week training programme (3 days a week) which covered, youth development, HIV and sexual health information, HIV counselling and testing, confidentiality, ethics, and research methods, study procedures and HIVST . Progress was evaluated using written and oral assessments to select 48 peer-navigators to work in pairs and implement the intervention in their areas. All packs also include written health promotion materials. Further details are provided under the study procedures.”

- The age limits for recruitment of AGYW requires some explanation.

RESPONSE: This has been justified in the manuscript as suggested and clarified above.

- Clarify the meaning of this “particularly the unreliability of HIVST on ART”

RESPONSE: This has been edited accordingly. This depicts that peer navigators should provide information suggested by the NDoH that HIVST can (rarely) give a false negative result in those who have been on longstanding ART.

- Coupon return is introduced with not previous mention/context

RESPONSE: Thank you. This has been edited accordingly.

- Specify “the same procedures as seeds”

RESPONSE: Further clarification has been provided in the manuscript.

- It is not clear what seeds are being expected to do

RESPONSE: Seeds are the first step in the respondent driven HIVST distribution. The seeds are expected to distribute their HIVST packs to their friends and encourage them to use the HIVST and visit the clinics. The targets are young women aged 18-24 years but not exclusively as stated in the manuscript.

- Arm 2

- Does recruitment= distribution of a pack?

RESPONSE: This has been clarified in the manuscript. Recruitment only begins at the time when participants visit our clinics for health services. We use the word ‘contact’ at the point of distribution.

- What sort of session will be undertaken? How long will it take? How will peer recruits be prepared for this?

RESPONSE: As with the response above, this is a brief verbal health promotion information where the peer navigators provide potential participants with information about health and social services available in their areas. Further details are provided under the study procedures. Participants do not need to prepare for this.

- The age limits for recruitment of AGYW requires some explanation.

RESPONSE: This has been justified in the manuscript as suggested.

- Clarify the meaning of this “particularly the unreliability of HIVST on ART”

RESPONSE: This has been edited accordingly. This depicts that peer navigators should provide information that HIVST will not give accurate result if someone is on ART.

- Arm 3

- What does a linkage information pack include?

RESPONSE: This includes study information, HIV and PrEP information, referral slips for the clinic. Further explanations are provided under the study procedures. (all arms have the same information – the only difference in the intervention arms is the addition of HIVST).

- “encourage” should be “encouraging”

RESPONSE: This has been edited accordingly. Thanks.

- What sort of session will be undertaken? How long will it take? How will peer recruits be prepared for this?

RESPONSE: See the answer above. This is a brief verbal health promotion where the peer navigators provide potential participants with information about health and social services available in their areas. Further details are provided under the study procedures. Participants do not need to prepare for this.

- Many of these things could be defined and explained just once to provide greater clarity.

RESPONSE: This has been done throughout the manuscript

Study oversight

- “convey” should be “convene”

RESPONSE: This has been edited accordingly.

- What was the role of the TAG in the study design?

RESPONSE: The TAG did not participate in the study design. The main task of the TAG is to monitor and supervise the progress of data collection, provide independent review of data collected during all cRCTs conducted under the STAR initiative, and assist investigators in disseminating results.

Study recruitment and procedures

- “fresh graduates” is not clear

RESPONSE: This has been edited accordingly.

- Provide some more details about recruitment of a training of peer navigators, what is the “several training and assessments” referred to?

RESPONSE: Further information has been provided. Between 6/2018-9/2018 participants underwent a 20-week training programme (3 days a week) which covered, youth development, HIV and sexual health information, HIV counselling and testing, confidentiality, ethics, and research methods, study procedures and HIVST. Progress was evaluated using written and oral assessments to select 48 peer-navigators to work in pairs and implement the intervention in their areas.

- 5000 young women overall? Ensure this is clearly stated throughout.

RESPONSE: We have edited the study setting and population section to clarify that the population of 18-30 year olds residing in the cluster RCT area is ~12,000 from which study participants will be recruited.

- How were the NDoH involved in the design of the intervention?

RESPONSE: The NDoH were involved in approving the study protocol and were one of the reasons the outcome is being measured in women aged 18-24 years. This has been edited accordingly.

- See above with regards to how the estimated targets were calculated?

RESPONSE: This has been edited accordingly. Further information is provided under the sample size section.

- What do the packs include and how are they colour coordinated and for what purpose?

RESPONSE: Please see figure 1 and study procedures for what is included in each pack per arm. The packs are colour coordinated so that if a participant links to clinic without their barcoded referral slip we can ascertain the arm of the study by asking them the colour of their pack / referral slip. This has worked well in the pilot to differentiate each arm. Further clarifications have been provided.

- “Scanned in”?

RESPONSE: This has been edited accordingly.

- Who are the participants being referred to in the first paragraph?

RESPONSE: We have clarified the difference between study recruitment and study enrolment.

- “The information is relevant”- what is the information that this is referring to, data collected in the questionnaire?

RESPONSE: The brief questionnaire data.

- How is the collection of process evaluation data collected in a different way from the data collected from participants?

RESPONSE: Process evaluation data collection is different from the clinical data that is being collected from participants when they visit the clinic. Process evaluation data seeks to evaluate the reach, design acceptability and feasibility of the study among others.

- How are the participants in paragraph 1 different from those in paragraph 2.

RESPONSE: This has been edited accordingly. They are not different.

- Will individuals enrolled in the facilities be enrolled regardless of interaction with peer recruiters in any arm? The eligibility criteria are not clear.

RESPONSE: Both walk-ins and study participants aged 18-30 years are eligible for receiving services from the clinic. However, only those who have been referred through one of the three arms, either identified through the barcoded referral slip, or following a brief screening questionnaire that has simple questions to ascertain if they were referred through by any of the arms, i.e. receiving any of the three colour coded packs/referral slips, or contact with a named peer navigator, or referral through peer network. Further clarifications have been provided.

- Do those enrolled at facilities complete any form of data collection process more than the screening?

RESPONSE: Yes. The clinical staff will ask them some eligibility questions before screening.

- The start of paragraph 3 fails to make clear whether this will be all visitors to the clinic or eligible visitors?

RESPONSE: This has been edited accordingly. See response above.

- If all clinics don't all offer ART and PrEP how will referrals be made and information about uptake be linked?

RESPONSE: Our institute has nurses and/or clinical research assistants in all the clinics in the study sites. They will refer participants to our clinics (including mobile clinics) that provide PrEP service. They will provide ART within the clinics since all the clinics provide ART service.

- Possibly split this section (final two paragraphs) to deal with HIV+ and HIV- separately to make sure processes are more clearly described.

RESPONSE: Noted with thanks. Further clarifications have been provided.

Inclusion and exclusion criteria

- May be more sensible to incorporate in recruitment and doing this by phase for greater clarity.
- There is limited information about or justification for required/targeted sample size for the study.

RESPONSE: We agree and have moved this above the section on recruitment and enrolment.

- The primary and secondary outcome as noted in the tables are not completely clear- only defined two pages after this section.

RESPONSE: Further clarifications have been provided on primary and secondary outcomes and we have moved them further up the manuscript as it helps clarify the eligibility, recruitment and enrolment and data collection.

Randomisation

- What is the PIPSA?

RESPONSE: Population Intervention Platform – we have removed this throughout and used demographic surveillance which provides greater clarity.

- What were the allocation restrictions? How do allocations map onto areas?

RESPONSE: Location (rural versus urban), HIV testing prevalence and uptake of DREAMS combination HIV prevention by adolescent girls and young women

- What is a PN?

RESPONSE: Peer navigators

- This section requires reworking for clarity- it is quite confusing.

RESPONSE: Further clarifications have been provided.

Sample size

- This needs to be discussed much earlier to help with clarity of process and flow.
- What is the estimated population size per area?

RESPONSE: We have clarified the study setting and population earlier and here we are explaining the sample size calculation. As we have said, the estimated population of eligible women aged 18-24 years is at least 200 per unit of randomization (i.e. each pair of peer navigator working area) and based on our pilot work we estimate that by working ~1000 hours per cluster area they will be able to reach the target population in their study area.

Outcomes

- The specific outcomes should be possibly discussed and presented alongside the study aims, the tools and analyses are better placed in an analysis section.

RESPONSE: This has been moved as suggested.

- What is the justification for the interim analysis? This should be placed within data collection.

RESPONSE: This was designed as a pragmatic trial that would have policy implications and so we have been asked to conduct a pre-planned interim analysis in that if there was a large effect it could be fed into policy.

- The primary outcome should be rewritten for more clarity. Difference with what? Formatting changes in this section.

RESPONSE: Further clarifications have been provided in the manuscript.

“The primary outcome is the difference in linkage rate between arms. Linkage rate is defined as the number of women (18-24 years) per peer-navigator month of outreach work (/pnm) who linked to clinic-based PrEP eligibility screening or started ART, based on HIV-status, within 90 days of referral. The rate is defined as the number of linkages per month of peer navigator outreach activities. The numerator is defined as the number of young women aged 18-24 years who attend clinic for confirmatory HIV testing, PrEP counselling or ART, following HIV-ST distribution or peer navigator referral to HIV testing, treatment and prevention services. The denominator for intention-to-treat analysis (ITT) is the entire time (study duration) spent by peer navigators doing their peer outreach work. For the on-treatment analysis, we will use the actual time spent by peer navigators on distributing packs in each arm. The time worked by each peer navigator will be combined to get the total time per pair of peer navigator. The difference in rate of linkage between the study arms will be calculated - incentivised HIVST delivery through peer network and direct distribution of HIVST will be compared to standard of care.”

Process evaluation

- If possible include all methods relating to this in one section.

RESPONSE: Okay. Thank you.

- Why will patterns of recruitment be considered in the evaluation?

RESPONSE: Purposive sampling will be employed to recruit participants for the process evaluation.

- How will unintended consequences and ethical issues be assessed?

RESPONSE: We rely on the study participants, community engagement units and community advisory boards, the hotline, as well as the peer navigators (during debriefings), clinic staff and the process evaluation to reveal any unintended consequences and social harms, including cases of coercive testing, under-age testing, inadvertent disclosure of HIV status, undue emotional/mental strain, or unexpected social harms in relation to HIV self-testing. These are reported through our adverse event monitoring form and are reviewed by the principal investigator and project management team. All AE and SAE are reported to the ethics committees and the study advisory group.

- How will reach will assessed?

RESPONSE: This will be calculated by the total number of young women contacted or reached by peer navigators over the six-month peer navigators' outreach.

- The use of and application of the MRC guidelines are not clear.

RESPONSE: This is an illustrative guideline. Further clarification has been provided.

Data collection

- Is the voucher the same as the coupon?

RESPONSE: Yes, but coupon has been used throughout the manuscript for consistency.

- Is the survey the same as the service recipient questionnaire?

RESPONSE: Yes

- How will the sample for IDIs be selected and based on what criteria?

RESPONSE: Participants will be selected purposively across the study sites. The majority of the participants will be those who were exposed to the peer intervention.

- What will issues/topics will the IDIs cover?

RESPONSE: This has been addressed accordingly in the manuscript.

Adverse events

- Seems logical for this to be addressed along with other ethical issues.

RESPONSE: Noted with thanks

- How will any instances of AE relating to HIVST be picked up by the study given these are largely circulated and used within the community?

RESPONSE: See the points above .

- Community engagement units and the telephone hotline are not described nor explained/given context.

RESPONSE: Further clarifications have been provided.

Analysis

- Link more clearly with outcomes

RESPONSE: Noted with thanks.

- Describe the qualitative analysis using an accepted method.

RESPONSE: This has been revised accordingly.

Discussion

- The argument about the intervention being derived from theory is not supported by the theory presented in the paper.

RESPONSE: We are yet to discuss the findings of the trial. The data emanating from the data will be analysed using appropriate theoretical model.

- The discussion presents a number of AEs, social harms and potential negative outcomes not addressed in the AE or ethics section- it is unclear how the study will be able to collect data on these.

RESPONSE: This has been addressed under Adverse Events section in the manuscript.

- Process issues including referrals and colour coding are not adequately discussed within the methods.

RESPONSE: Further clarifications have been provided.

Reviewer 2:
General comments

=====

The authors have written a clear, well-motivated protocol for a 3-arm, randomized controlled trial of interventions to increase uptake of HIV testing and linkage to prevention & care through use of peer-delivery HIV self-test kits and education. The protocol clearly outlines the study design and rationale, study methods for recruitment, enrollment, and outcomes to be measured and compared. Importantly, the authors also include qualitative methodology to further investigate the outcomes beyond numeric percentages, as well as cost-effectiveness studies to assist with translating protocol outcomes into policy recommendations.

Authors' Responses to Reviewer 2:

RESPONSE: Thank you very much for your positive feedback on the relevance of this paper.

Reviewer 3:
General comments

=====

This is a very well written manuscript. I have minor comments:

Authors' Responses to Reviewer 2:

I am not sure why there were two inclusion/exclusion criteria one for i.e. the recruitment by Peer Navigators and/or Seeds to receive HIVST packs or clinical referral slips which was specified as 18-30 years; while, primary and the secondary outcomes will be ascertained among 18 to 24 years old

RESPONSE:

The primary outcome is being measured in adolescent girls and young women aged 18-24 because adolescent girls and young women (aged 15-24 years) have been defined by the South African NDoH as a key population for targeting HIV prevention, including PrEP, due to ongoing high HIV incidence. National guideline has suggested that HIVST in those under 18 years should be supervised by a healthcare worker and for this reason we had to limit our primary outcome based on eligibility for PrEP if negative to 18-24 years. However, our peer navigators programme is working with all young people under age 30 years to increase uptake of HIV testing, care and prevention. Therefore, although the community promotion of HIV testing and linkage to care was provided to men and women aged 18-30 years, we have focused on measuring the primary outcome of linkage to PrEP in young women aged 18-24 years.

We have reordered the background to ensure that the persistent challenge of high incidence of HIV in young women aged 15-24 years is clear (paragraph 1); explain the NDoH guidance in HCW supported testing in those <18 (para 3) and NDoH guidelines on PrEP targeting (para 4); and then in

last paragraph of the background we have clarified why the intervention is delivered to young people aged 18-30 years but the outcome of linkage to PrEP is measured in those aged 18-24 years.

Page 12: is there any reason that the lines 9-10 used bold/bigger format

RESPONSE: No and this has been corrected

Statistical analysis: It is not clear how the primary and secondary outcomes will be analysed?

- Investigators stated that they will be using t-test and 95% CIs. What will be compared using the t-test? Cluster-level summaries?

RESPONSE: Yes, these are cluster level summaries – we have made changes to the stats (see below) to clarify

- On Page 14: what does “standard methods” means?

RESPONSE: Standard methods for analysing cluster-randomised trials with small numbers of clusters – Since the number of clusters are small, the effect of the intervention will be estimated using a two-stage approach based on cluster-level summaries [ref Hayes & Moulton]. The cluster-level approach, although less statistically efficient than methods based on individual level regression, is more robust when there are a relatively small number of clusters. All analyses will be performed using STATA version 15 (StataCorp LP, College Station, Texas USA).

-Investigators do not plan any adjusted analysis; however, it is usually a custom to suggest an adjusted analysis in case there was a significant imbalance between the study arms. Is there a reason that the investigators do not expect any imbalance?

RESPONSE: This was an error sorry. A rate ratio adjusting for substantial covariate imbalance at baseline will also be calculated, using a two-stage process; all covariates will be pre-specified in the analysis plan. To identify covariates for adjustment, baseline characteristics of each arm will be presented, and the size of the difference of covariates known to be associated with the outcome will be assessed quantitatively.

- Then, later in statistical analysis section (page 14 lines 24-27), it was stated that “Substantial differences will be identified by comparing frequencies or means of variables known to be associated with the primary outcome. These will be assessed by investigators without the use of statistical tests.” My question is: How the substantial differences will be assessed without the use of statistical tests? Do they mean “no p-values will be presented”

RESPONSE: Since the clusters have been randomly allocated to the arms, any differences that exist must be due to chance, therefore the null hypothesis is known to be true and statistical tests are not appropriate. We will present the baseline characteristics of each arm, so that the size of the differences can be assessed quantitatively, and a decision made on the need for adjustment. To identify covariates for adjustment, baseline characteristics of each arm will be presented, and the size of the difference of covariates known to be associated with the outcome will be assessed quantitatively.

We have rewritten the analysis plan which we hope is now clear:

“The analysis of primary outcome follows an intention-to-treat (ITT) and per-protocol approaches. The analysis of secondary outcomes will be undertaken using per-protocol approach only. The primary outcome compares the difference between the rate of linkage of 18-24-year-old women to HIV confirmatory HIV testing, ART (if HIV positive) or PrEP counselling (if HIV negative). The rate is defined as the number of linkages per month of peer navigator outreach activities. The numerator is defined as the number of young women aged 18-24 who attend clinic for confirmatory HIV testing, PrEP counselling or ART, following HIV-ST distribution or peer navigator referral to HIV testing, treatment and prevention services. The denominator for intention-to-treat analysis (ITT) is the entire time (study duration) spent by peer navigators doing their peer outreach work. For the on-treatment analysis, we will use the actual time spent by peer navigators on distributing packs in each arm. The time worked by each peer navigator will be combined to get the total time per pair of peer navigator. The difference in rate of linkage between the study arms will be calculated - incentivised HIVST delivery through peer network and direct distribution of HIVST will be compared to standard of care. Since we randomised the pairs of peer navigator (clusters), the rate of linkage will be calculated for each pair of peer navigator using aggregate data for each cluster. Since the number of clusters are small, the effect of the intervention will be estimated using a two-stage approach based on cluster-level summaries [ref Hayes & Moulton]. The cluster-level approach, although less statistically efficient than methods based on individual level regression, is more robust when there are a relatively small number of clusters. All analyses will be performed using STATA version 15 (StataCorp LP, College Station, Texas USA).

Cluster-level linkage rates will be calculated and used to estimate the unadjusted rate ratio and its 95% confidence interval for the effect of each intervention arm compared with the standard of care; the mean difference in linkage rates between each arm and standard of care and against each-other will be assessed using a t-test. A rate ratio adjusting for substantial covariate imbalance at baseline will also be calculated, using a two-stage process; all covariates will be pre-specified in the analysis plan. To identify covariates for adjustment, baseline characteristics of each arm will be presented, and the size of the difference of covariates known to be associated with the outcome will be assessed quantitatively.

As part of the exploratory analysis, we will perform a (i) subgroup analysis by gender and area and (ii) two intervention arms will be compared to one another (incentivised HIVST delivery through peer network approach will also be compared to direct distribution of HIVST approach). To expand on this, the data from the client survey captured on REDCap dashboard will be exported into STATA, cleaned and analysed. All reporting will conform to CONSORT guidance for cluster randomised trials. “

VERSION 2 – REVIEW

REVIEWER	Lucia Knight School of Public Health, UWC, South Africa
REVIEW RETURNED	19-Nov-2019
GENERAL COMMENTS	The requested changes have been comprehensively addressed.
REVIEWER	Handan Wand University of New South Wales, Australia
REVIEW RETURNED	17-Nov-2019
GENERAL COMMENTS	This is a very well written study. I think the authors did a great job for responding the comments raised by the reviewers. I have no further comments.